# Measurement-induced criticality in extended and long-range unitary circuits

**Shraddha Sharma[1⋆], Xhek Turkeshi[1,2,3], Rosario Fazio[1,4] and Marcello Dalmonte[1,2]**

**1** The Abdus Salam International Center for Theoretical Physics (ICTP),
Strada Costiera 11, 34151 Trieste, Italy
**2** International School of Advanced Studies (SISSA), via Bonomea 265, 34136 Trieste, Italy
**3** JEIP, USR 3573 CNRS, Collége de France, PSL Research University,
11 Place Marcelin Berthelot, 75321 Paris Cedex 05, France
**4** Dipartimento di Fisica, Università di Napoli Federico II,
Monte S. Angelo, I-80126 Napoli, Italy

⋆ ssharma@ictp.it

## Abstract

We explore the dynamical phases of unitary Clifford circuits with variable-range interactions, coupled to a monitoring environment. We investigate two classes of models, distinguished by the action of the unitary gates, which either are organized in clusters of finite-range two-body gates, or are pair-wise interactions randomly distributed throughout the system with a power-law distribution. We find the range of the interactions plays a key role in characterizing both phases and their measurement-induced transitions. For the cluster unitary gates we find a transition between a phase with volume-law scaling of the entanglement entropy and a phase with area-law entanglement entropy. Our results indicate that the universality class of the phase transition is compatible to that of short range hybrid Clifford circuits. Oppositely, in the case of power-law distributed gates, we find the universality class of the phase transition changes continuously with the parameter controlling the range of interactions. In particular, for intermediate values of the control parameter, we find a non-conformal critical line which separates a phase with volume-law scaling of the entanglement entropy from one with sub-extensive scaling. Within this region, we find the entanglement entropy and the logarithmic negativity present a cross-over from a phase with algebraic growth of entanglement with system size, and an area-law phase.

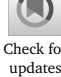

# 1   Introduction

In the past years, the interplay between unitary evolution and quantum measurement has been the focus of intensive studies in many-body physics [1–57]. The resulting dynamics is that of a stochastic trajectory in the Hilbert space, determined both by the system interactions, and by the measurement outcomes of the environment probing. While the former drive the system to explore large regions of the Hilbert space, constrained only by the conservation laws of the model, the latter localize the wavefunction as a result of the collapsing nature of the measurements. The consequences of this competition are manifested in quantities that are not directly captured by the averaged density matrix of the stationary state, such as quantum information measures: by varying the rate or strength of the measurements, the system undergoes a phase transition, whose universality class is related both to the system unitary dynamics and to the measurement protocol used. Differences among these phases are equally well captured by other observables, provided they are non-linear in the wave-function [2, 24, 36–44, 52].

A "minimal model" for the study of the transition is that of hybrid random circuits. These systems comprise random unitary gates drawn from a unitary ensemble, and measurement gates which pervade a discrete space-time geometry with an *ab-initio* specified protocol. Presently, two ensembles have been thoroughly considered, Haar unitary ensemble and Clifford unitary ensemble, and mostly with short-range interactions. The statistical properties of the Haar distribution allow for exact mapping of the hybrid random circuit to classical statistical mechanics models [7, 48, 58–63], where analytical results were obtained in the limit of large qudit dimension. Instead, stabilizer states and error correcting methods allow efficient classical simulations for Clifford hybrid random circuits, provided the measuring gates are projectors on the Pauli group [1, 2, 64–68]. In particular, Clifford circuits provide a viable path to understand the role of the dimensionality [31, 33, 34], topological features [11, 11, 39], and the nature of the critical point [15, 44, 69].

Recently these hybrid system have been implemented experimentally in trapped-ion hardware [46], where other experimental realizations have been proposed in Ref. [35, 42]. Crucially, trapped ions naturally allow to introduce controllable long-range interactions [70], hence with a unitary evolution that in general favours entanglement between arbitrary far spins of the system. This detail is far from being irrelevant: for instance in isolated Hamiltonian systems, correlations might be subjected to considerably different Lieb-Robinson bounds when compared to short-range systems [71], whereas entanglement generation can be related

to an emergent semiclassical picture, that breaks down when the controllable interaction is sufficiently *short ranged* (see Ref. [72–74] and references therein for a theoretical discussion, and Ref. [75–80] and references therein for experimental implementation and discussion).

In the context of hybrid systems, measurement-induced phase transitions (MIPT) have been considered in Ref. [7] for Haar long-range hybrid circuits, in Ref. [49,50] for system with free fermion long-range Hamiltonian, and in Ref. [51] for Clifford hybrid circuits. While the various protocols differ in several aspects (some of which we discuss in more detail below), a leitmotif of long-range circuits is that those can support critical scenarios that are sharply distinct to short-range systems, in full analogy with what happens for equilibrium critical behavior [81]. Two specific features that have been emphasized in these works are the loss of Lorentz invariance at MIPT, as well as the presence of phases displaying sub-volumetric entanglement entropy scaling.

In line with these recent developments, in this paper we explore the dynamical phase diagram of the system at variable range of interaction. Specifically we implement two different protocol in Clifford hybrid random circuits (illustrated schematically in Fig. 1b), aiming in exploring the effect of finite range correlating gates (soft-shoulder potential) and power-law decaying interactions, which are suitable for the theoretical description of trapped-ion platforms.

In the first one – dubbed cluster hybrid random circuit (CHRC), each layer of the unitary dynamics is composed of clusters extending over $M$ sites, each one of which is build from elementary two-body gates at a finite range $r \leq M$.

In the second – which we call long-range hybrid random circuit (LRHRC), two-body random unitary gates may extend throughout the system, and are drawn from a probability distribution $P(r) \sim r^{-\alpha}$. Compared to Ref. [51], we do not fix the number of unitary gates per time step, but rather consider a stochastic number of unitaries. The normalization of $P(r)$ acts as a Kac normalization [72–74,82], as fix for the average number of two-body long-range unitary to be extensive.

In both scenarios we perform projective measurements for the on-site spin polarization. To diagnose the phases and the measurement-induced transition we consider different quantum information measures: the bipartite entanglement entropy, the bipartite and tripartite mutual information, and the entanglement negativity.

Our main goals are two. The first one is to shine further light on the interplay between measurement and non-local interactions, complementing the picture discussed so far, by considering two scenarios where non-locality is varied in a controlled manner. The second one consists of an in depth characterization of the entanglement properties - in particular, as quantified by the violation of the positive-partial transpose condition [83,84] - in monitored long-range systems.

We find that in the case of CHRC, the model exhibits a sharp transition between an error correcting phase and a quantum Zeno phase, characterized respectively by extensive/constant scaling with system size of the stationary state entanglement entropy. Our analysis suggests the fixed point is compatible with the one of short-range Clifford circuits. Importantly, we identify the transition for a finite-measurement rate up to a certain cluster size. At larger cluster range, the transition point approaches the fine tuning point $p \to 1$, thus highlighting the presence of only a volume-law error correcting phase.

In the case of LRHRC, we identify a line of critical points varying the range of interactions. In particular, we find three regimes characterized by the exponent $\alpha$ controlling the interaction range. For $\alpha \geq 3$ the system belongs to the same universality class of the short-range Clifford circuits, whereas $1 \leq \alpha < 3$ displays non-conformal critical points, as signaled by the presence of a non-unity dynamical critical exponent $z$, and by a non-zero scaling dimension $\gamma_{\mathcal{Q}}$ for the order parameters $\mathcal{Q}$. In addition, we find an intermediate phase $1 \leq \alpha \lesssim 2$ characterized by

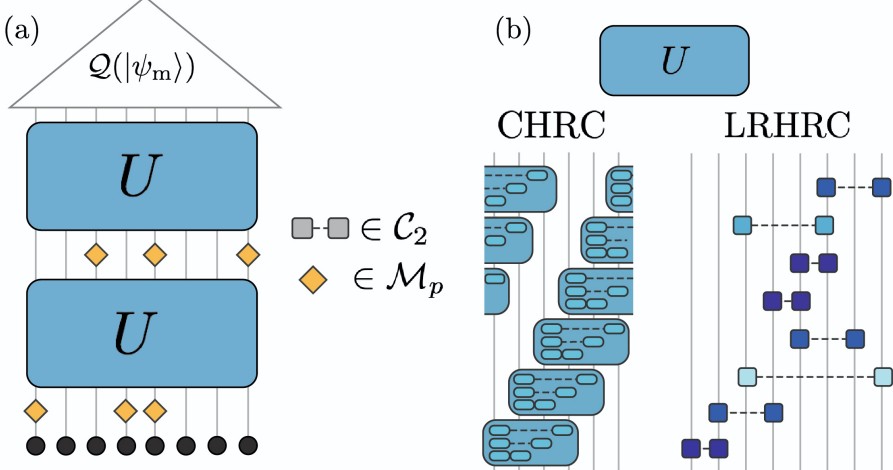

Figure 1: (a) The hybrid random circuit is composed of measurement layers interspersed with unitary layers. The circuit is initialized in a product state of spin variables, and, at the stationary limit, the quantity of interest $\mathcal{Q}$ is evaluated for the quantum trajectory $|\psi_{\mathbf{m}}\rangle$, where $\mathbf{m}$ is a collective index that specifies a trajectory. The unitary evolution is build upon two body random Clifford gates $U_{i,j} \in \mathcal{C}_2$, whereas the projective measurement are applied on each site with probability $p$. If a measurement occurs, the collapsing operator is chosen from $\mathcal{M}_p = \{(1+Z)/2, (1-Z)/2\}$. (b) Unitary layer for CHRC and LRHRC protocols. For the CHRC, we depict a periodic system of $L = 6$ sites: the collective unitary layer is composed of $L$ cluster unitaries acting on $M$ spins (big rectangles), which are build out of $M-1$ two-body random circuit (small rectangles). In the above illustration $M = 4$. (See Eq. (3) for the general definition). For the LRHRC the constituting two-body unitary gates act on $i, j$ drawn from the lattice with probability $P(r = |i-j|) \propto 1/|j-i|^{\alpha}$. The normalization of the probability distribution $P(r)$ imposes an effective Kac normalization on the system. In the cartoon in (b-Right), the different colors reflect different probability density of the associated indices $(i, j)$.

a sub-extensive (algebraic) scaling with system sizes of the entanglement entropy and of the negativity between antipodal regions. For $\alpha \gtrsim 2$ both these quantities exhibit a crossover to a phase where the entanglement entropy is constant (area-law) and the negativity between antipodal regions decreases with a power law with system sizes. The phenomenology we observe is consistent with the one reported in Ref. [51], suggesting that, at least for the case of Clifford circuits, the measurement induced transition is governed by the same underlying, $\alpha$-dependent theory despite the protocols being different.

The paper is structured as follows. In Sec. 2 we introduce the models of interest, and the relevant entanglement measures are defined in Sec. 3. In Sec. 4, we present the numerical results: in Sec. 4.1 we analyse the CHRC model, while in Sec. 4.2 the LRHRC model. We discuss our findings and conclude the paper in Sec. 5.

## 2 Model and protocols

We consider a one-dimensional lattice of $L$ spin-1/2 qubits, initialized in a product state $|\psi_0\rangle$, and let it evolve for a time $T$ via a hybrid Clifford circuit. At each time step the system first is probed by the environment, which measures independently each site with a probability rate

$p$, and then unitarily evolved. (A pictorial summary is presented in Fig. 1(a)). The state thus follows a quantum trajectory

$$|\psi_{\mathbf{m}}\rangle \equiv \frac{K_{\mathbf{m}}}{||K_{\mathbf{m}}|\psi_0\rangle||}|\psi_0\rangle, \tag{1}$$

where $K_{\mathbf{m}}$ is the realization of the circuit, the index $\mathbf{m}$ contains the measurement position record, the choice of unitaries and the measurement outcomes, and the denominator is a renormalization that introduces a non-linearity in the quantum evolution.

When the measurement is performed, the Born rule determines the measurement outcome and hence the post-measurement state. We consider the measurements over the magnetic polarization $\mathcal{M}_p = \{(1+Z)/2, (1-Z)/2\}$[1]. The action of measurement then gives

$$|\psi\rangle \mapsto \frac{P_j^{\pm}|\psi\rangle}{||P_j^{\pm}|\psi\rangle||}, \qquad P_j^{\pm} = \frac{1 \pm Z_j}{2}. \tag{2}$$

The unitary layer determines the specific protocol at hand: either we consider the pattern of a soft-shoulder potential (Sec. 2.1), or we consider arbitrary far, power-law distributed interactions (Sec. 2.2). In both cases, the elementary building blocks are two-body random Clifford gates $U_{i,j} \in \mathcal{C}_2$. (A brief discussion on the stabilizer formalism, Clifford group and on the efficient numerical implementation based on the Gottesmann-Knill theorem are given in the Appendix).

## 2.1 Cluster hybrid random circuit (CHRC)

The unitary evolution on the system is a layer build on $L$ $M$-body cluster unitary gates $\mathcal{U}_{\{i,\ldots,i+M-1\},t}$, each of which is build stacking two body unitary gates $U_{i,j,t}$ with progressively increasing range $|i-j| = 1, \ldots, M-1$[2] (See Fig. 1(b) for a pictorial representation). Concretely, the unitary at time-step (circuit depth) $t$ is given by

$$U(t) = \prod_{i=1}^{L} \mathcal{U}_{\{i,i+1,\ldots,i+M-1\},t} \equiv \mathcal{U}_{\{L,1,\ldots,M-1\},t}\mathcal{U}_{\{L-1,L,1\ldots,M-2\},t}\cdots\mathcal{U}_{\{1,\ldots,M\},t},$$

$$\mathcal{U}_{\{i,\ldots,i+M-1\},t} = \prod_{r=1}^{M-1} U_{i,i+r,t} \equiv U_{i,i+M-1,t}\cdots U_{i,i+2,t}U_{i,i+1,t}. \tag{3}$$

In the above equations, periodic boundary conditions ($L + i = i$ for any $i = 1,\ldots,L$) are inferred, and for clarity on the ordering of the unitaries, we explicitly wrote the expansion of the product. The unitary gates $U(t)$ are laid out in a manner that mimics a soft-shoulder potential extending over $M$ sites.

As the range of unitary evolution, characterized by the cluster size $M$, is altered, the information scrambling properties of the circuit $K_{\mathbf{m}}$ are varied. In the following we are going to discuss the instance of even $M$, and $L$ multiple of $M$, to avoid commensurability effects due to the boundary condition.

---

[1] We use the convention $X$, $Y$, and $Z$ to indicate, respectively, the $\sigma^x$, $\sigma^y$ and $\sigma^z$ Pauli matrices.

[2] Throughout the paper, the subscript denote the only non-trivial action. That is, denoting the cardinality of the set $Y$ as $|Y|$, any operator $O_{\{i\in I\}} \equiv O_{i_1,i_2,\ldots,i_{|I|}}$ acts non-trivially only on the qubits $\{i \in I\}$. Rephrasing, the basis representation of each of the operator is given by $\langle\sigma_1,\sigma_2,\ldots,\sigma_L|O_{\{i\in I\}}|\tau_1,\tau_2,\ldots,\tau_L\rangle = \left(\prod_{i\notin I}\delta_{\sigma_i,\tau_i}\right)(O_{\{i\in I\}})_{\tau_{i_1}\tau_{i_2}\ldots\tau_{i_{|I|}}}^{\sigma_{i_1}\sigma_{i_2}\ldots\sigma_{i_{|I|}}}$.

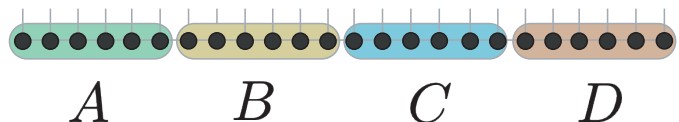

Figure 2: Partitions considered for the stationary state observables. Each part $\Xi \in A, B, C, D$ has equal length given by $|\Xi| = L/4$. For the half-system size entropy we consider $B \cup C$, whereas for the other quantities of interest, the partition is explicitly stated. Boundary conditions are periodic.

## 2.2 Long-range hybrid random circuit (LRHRC)

In this case, the unitary matrix $U(t)$ is build as a product of 2 qubit gates $U_{i,j,t}$, where $i$ and $j$ are distributed in such a way that

$$P(|i-j|) = \frac{1}{\mathcal{N}} \frac{1}{|i-j|^\alpha}, \qquad \mathcal{N} = \sum_{r=1}^{L-1} \frac{1}{r^\alpha}. \qquad (4)$$

The normalization $\mathcal{N}$ in Eq. (4) is a Kac normalization that guarantees that the average number of two-body unitary gates acting on each unitary layer scales as $\propto L$. Hence, with the above prescription, and given the set $I_t$ of pairs $(i, j)$ extracted with Eq. (4) at time $t$, we have $U(t) = \prod_{(i,j) \in I_t} U_{i,j,t}$.

We point out that a similar model has already been considered in Ref. [51], where the number of two-qubit gates was fixed to be $L$, and were entanglement entropy, mutual information, and average purification time in the context of purification transition were considered.

While it is important to stress that Clifford unitary gates are not necessarily related to an underlying unitary dynamics, we note that the typical microscopic realizations of both trotterized and analog dynamics typically rely on the same process - one example being phonon-mediated spin interactions in trapped ion systems [70]. In this context, the two types of dynamics we discuss are close cousins to Hamiltonian with soft-shoulder potentials (Eq. 3), and spin-models with power-law decaying couplings (Eq. 4), respectively.

We conclude this section by specifying the analytical expression for the overall circuit. We define the set of measurement positions at time $t$ as $I_t^{\text{meas}}$, and the overall measurement operator at time $t$ as $\Upsilon(t) = \prod_{i \in I_t^{\text{meas}}} P_i^{q_i}$, where $q_i = \pm$ is fixed by the Born rule. Then $K_{\mathbf{m}} = \prod_{t=1}^{T} \Upsilon(t) U(t)$.

## 3 Observables

Before specifying the quantities of interest, it is instructive to spell out the difference between the quantum trajectories described by Eq. (1) and the average state

$$\overline{\rho} = \mathbb{E}_{\mathbf{m}}(|\psi_{\mathbf{m}}\rangle\langle\psi_{\mathbf{m}}| \times \mathcal{P}_{\mathbf{m}}) = \mathbb{E}_{\mathbf{m}}(K_{\mathbf{m}}|\psi_0\rangle\langle\psi_0|K_{\mathbf{m}}^\dagger) \equiv \Phi(|\psi_0\rangle\langle\psi_0|). \qquad (5)$$

In the above equation, the average $\mathbb{E}_{\mathbf{m}}$ includes three terms: the average over the measurement locations, the average over the random Clifford gates, and the average over the measurement outcomes, and $\mathcal{P}_{\mathbf{m}} = |||\psi_{\mathbf{m}}\rangle||^2$ is the probability density of the circuit realization. As clear in Eq. (5), the average state $\overline{\rho}$ evolves according to the quantum channel $\Phi$ whose precise form is fixed by the details of the circuit [64]. For an operator which is linear in the density matrix, we have (see for instance Ref. [58])

$$\overline{O} \equiv \mathbb{E}_{\mathbf{m}}(\langle\psi_{\mathbf{m}}|O|\psi_{\mathbf{m}}\rangle \times |||\psi_{\mathbf{m}}\rangle||) = \mathbb{E}_{\mathbf{m}}(\langle\psi_0|K_{\mathbf{m}}^\dagger O K_{\mathbf{m}}|\psi_0\rangle)$$
$$= \text{tr}(O\Phi(|\psi_0\rangle\langle\psi_0|)) = \text{tr}(O\overline{\rho}). \qquad (6)$$

Hence, this class of observables does not distinguish between the average over trajectories and the expectation value over the average state. *En passant*, we note that this justifies the Monte Carlo methods where the unravelling quantum trajectories are used to compute the dynamics of expectation values in dissipative quantum systems [85].

Instead, for operators $F$ that are non-linear in the density matrix, the expectation value and the trajectory average do not commute

$$\overline{F} \equiv \mathbb{E}_{\mathbf{m}}(F(|\psi_{\mathbf{m}}\rangle\langle\psi_{\mathbf{m}}|) \times ||\psi_{\mathbf{m}}\rangle||) \neq \text{tr}(F(\overline{\rho})).\tag{7}$$

A simple example is given by the purity $F(\rho) = \rho^2$: in this case, it is clear that $\overline{F} = 1$, whereas $\text{tr}(F(\overline{\rho})) = \text{tr}(\overline{\rho}^2) < 1$, since the state is mixed.

The above discussion justifies the use of entanglement and information measures, which are in general non-linear functionals over the state, as suitable candidates to identify the transition and characterize these dynamical phases. In particular, for stabilizer states and hence for Clifford circuits, these quantities can be computed in polynomial computational resources [17, 18, 67, 68]. Throughout this paper, we consider bipartite entanglement entropy, bipartite and tripartite mutual information and entanglement negativity as observables of interest. We detail in the Appendix how these quantities are computed within the stabilizer formalism.

**Entanglement entropy (EE)**   Given a bipartition of the system $A \cup B$, and a quantum state $|\psi\rangle$ the (bipartite) entanglement is contained in the density matrix $\rho_A = \text{tr}_B|\psi\rangle\langle\psi|$, where the partial trace $\text{tr}_B$ involves only the degrees of freedom in $B$. The EE is then defined as

$$S_A = -\text{tr}_A(\rho_A \log_2 \rho_A).\tag{8}$$

This quantity is a measure of the Bell pairs that can be distilled over the state $|\psi\rangle$, and has been extensively considered in many-body physics as an indicator of phases and phase transitions.

**Bipartite (MI) and tripartite mutual information (TMI)**   Given a tripartition $A \cup B \cup C$, the correlations between the partition of $A$ and $C$ are not captured by the EE defined above. It is convenient to consider the MI between the $A$ and $C$, defined as

$$I_2(A : C) = S(A) + S(C) - S(A \cup C).\tag{9}$$

We also consider the TMI, as it has been shown that for short-range unitary circuits it provides significant improvement in the numerical data quality, as less affected by finite size corrections compared to the MI. This is defined given a quadripartition $A \cup B \cup C \cup D$ as

$$I_3(A : B : C) = S(A) + S(B) + S(C) - S(A \cup B) - S(A \cup C) - S(B \cup C) + S(A \cup B \cup C).\tag{10}$$

We note that both these quantities include classical correlations, hence they do not quantify the reciprocal entanglement between subparts.

**Entanglement negativity**   To circumvent the limits of the MI and TMI, we consider the entanglement (logarithmic) negativity [84]. To define this quantity, we introduce the partial transpose. We consider a tripartition $A \cup B \cup C$ and a pure state $|\psi\rangle$. Tracing out $B$, we obtain $\rho_{AC} = \text{tr}_B(|\psi\rangle\langle\psi|)$, whose matrix elements are

$$\langle\{\phi\}_A; \{\phi\}_C|\rho_{AC}|\{\phi'\}_A; \{\phi'\}_C\rangle.\tag{11}$$

The partial transpose $\Gamma_A$ over the partition $A$ is given by the matrix

$$\langle\{\phi\}_A; \{\phi\}_C|\rho_{AC}^{\Gamma_A}|\{\phi'\}_A; \{\phi'\}_C\rangle \equiv \langle\{\phi'\}_A; \{\phi\}_C|\rho_{AC}|\{\phi\}_A; \{\phi'\}_C\rangle.\tag{12}$$

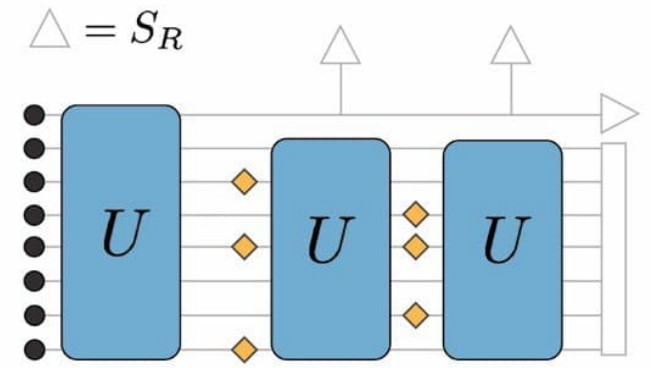

Figure 3: Scheme for the ancilla entanglement entropy. After a single time step fully entangling the ancilla qubit with the remaining of the system, the circuit is let evolve and the single qubit entropy $S_R$ is computed on the ancilla at each time step.

Then, the quantity

$$\mathcal{E}(A:C) = \log_2 ||\rho_{AC}^{\Gamma_A}||_1, \tag{13}$$

is an entanglement monotone with respect to the entanglement between $A$ and $C$. Note that such monotone is sensitive solely to entanglement related to the violation of the positive partial transpose condition [83]. In Eq. (13), $||O||_1 \equiv \sqrt{O^\dagger O}$ is the trace norm. An important bound is $\mathcal{E}(A:C) \leq I_2(A:B)/2$.

**Ancilla entanglement entropy**   In order to obtain the dynamical critical exponent, we couple an ancillary qubit R with a single site of the system [14,15]. (This coupling is induced by a layer of fully connected random unitary gates acting on the system and ancilla qubit R, shown in first step of Fig. 3.) The circuit then acts only on the system part, and we study the entanglement behavior of the ancilla EE $S_R$. We remark that $S_R$ acts as an order parameter for the steady state. In error-correcting phases its value is non-zero, whereas after a critical measurement rate, it is zero due to the dis-entangling effect of local measurements [15]. In this work, we opt to consider only the dynamical behavior of this quantity to extract the exponent $z$ as detailed below, as the stationary value of $S_R$ do not provide insights on the entanglement scaling with system size within the phases.

**Time evolution, stationary state and measurement-induced transition**   For each quantity $\mathcal{Q} \in \{S_A, I_2(A:C), I_3(A:B:C), \mathcal{E}(A:C), S_R\}$ and for each trajectory Eq. (1), we compute $\mathcal{Q}(\mathbf{m}) \equiv \mathcal{Q}(|\psi_{\mathbf{m}}\rangle\langle\psi_{\mathbf{m}}|)$. The main quantity of interest is then the conditional average over the trajectories

$$\mathcal{Q} \equiv \mathbb{E}_{\mathbf{m}}(\mathcal{Q}(\mathbf{m}) \times |||\psi_{\mathbf{m}}\rangle||). \tag{14}$$

To simplify the notation, in the following we omit the overline to denote the conditional average. With this end, we consider circuits $K_{\mathbf{m}}$ of increasing depth $0 \leq t \leq T$, with $T = 4L$ the maximum time, for which we checked the system reaches the stationary state. For the quantities $\mathcal{Q} = I_3, I_2, S_A, \mathcal{E}$ we consider the stationary values, which are informative of the entanglement content of the stationary state, whereas for the time evolution we study $\mathcal{Q} = S_R$. We identify the phase transition by performing a finite size scaling for the observables of interest $\mathcal{Q}$. For $\mathcal{Q} = I_3, I_2, S_A, \mathcal{E}$ we consider the scaling hypothesis

$$\mathcal{Q}(p \simeq p_c) = L^{\gamma_{\mathcal{Q}}} f_{\mathcal{Q}}((p - p_c)L^{1/\nu}), \tag{15}$$

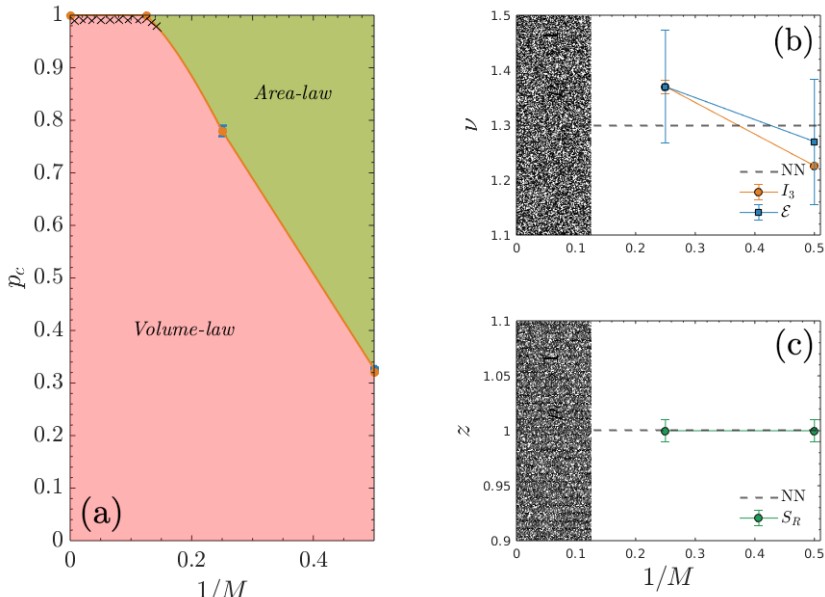

Figure 4: (a) The plot for the critical $p$ as a function of the range of unitaries $1/M$. The $p_c$ is extracted using both $I_3$ and $\mathcal{E}$ for comaprision. The region marked 'Volume-law' shows a length dependent scaling of EE, whereas, in the region marked 'Area-law' EE scales independent of $L$. Our numerics show $p_c$ compatible with 1 within error bars for $M > 6$ (graphically shown as the crossed region in the above plot). Correspondingly, the behavior of critical exponents (b) $\nu$ obtained using $I_3$ and $\mathcal{E}$ and (c) z extracted using $S_r$, as a function of $1/M$ is presented. The exponent $\nu$ changes as $1/M$ is changed whereas $z = 1$, within error bars. In both (b) and (c), the dotted line represents the nearest-neighbour values as obtained from the short ranged circuit.

where $\gamma_{\mathcal{Q}}$ is the scaling exponent of the quantity $\mathcal{Q}$, while $\xi = (p - p_c)L^{1/\nu}$ is the divering correlation length. For the observables of interest, $\gamma_{\mathcal{Q}} = 0$ for CFT critical point, whereas $\gamma_{\mathcal{Q}} \neq 0$ for non-conformal criticality. We retrieve the dynamical critical exponent $z$ by considering the entanglement of the ancilla qubit

$$S_{\mathrm{R}}(t; p_c) = f(t/L^z). \tag{16}$$

Notice that since we are not considering an extensive number of qubits, $S_{\mathrm{R}}$ need not to be renormalized via a scaling exponent. (A complementary discussion on purification dynamics is instead given in Ref. [51]).

# 4 Results

In this section we examine the (conditional averages) of the observables introduced in Sec. 3. For both the protocols in Sec. 2, we run simulations varying system size $L$, measurement rate $p$, and the control parameter $g$ ($g = M$ for CHRC, and $g = \alpha$ for LRHRC). Our findings show the correlation length $\xi$ is a function of the parameter $g$ controlling the range of interaction. Consistently, we extract the exponent $\nu$ and the dynamical critical exponent $z$. In Sec. 4.1 and in Sec. 4.2 we present our findings respectively for CHRC, and the LRHRC.

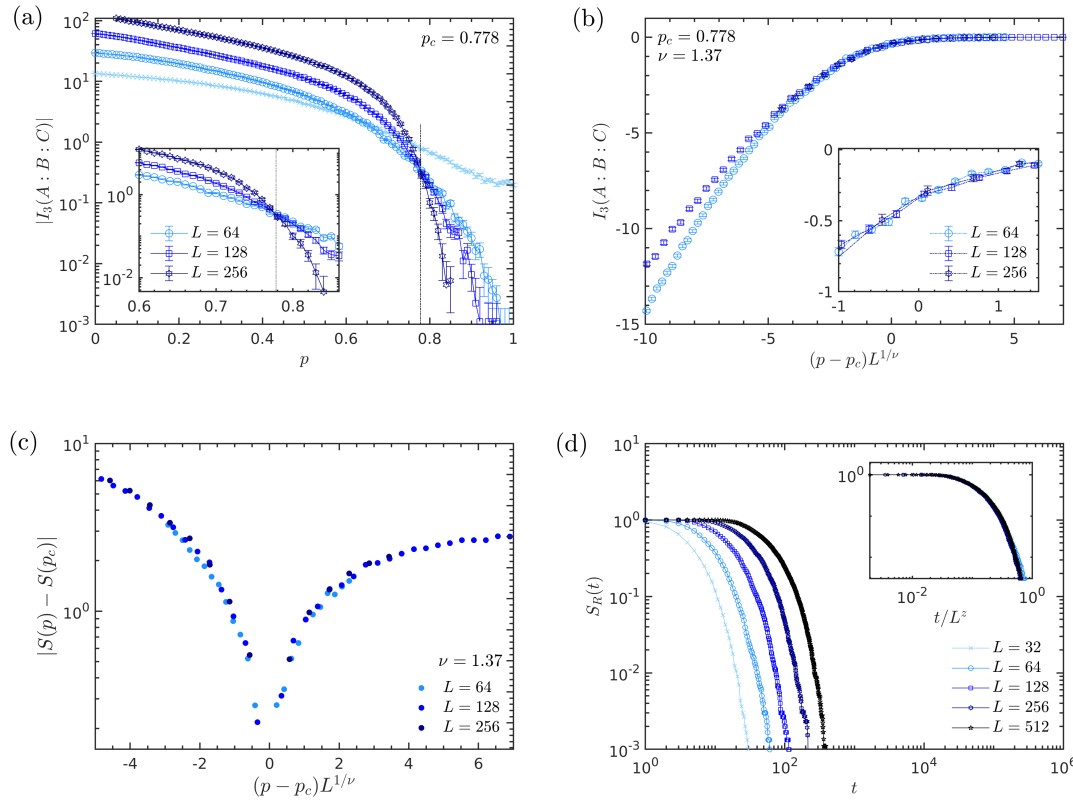

Figure 5: The behavior of relevant observables in order to obtain the critical exponents for $M = 4$. (a) TMI for larger system sizes show a crossing (dashed-dotted line) at critical $p_c = 0.778$, shown more clearly in the inset. (b) The collapse of TMI for $\nu = 1.37$ and $p_c = 0.778$. (c) The collapse behavior of EE obtained at $p_c$ for $\nu = 1.37$. (d) The exponent $z = 1$ extracted from the scaling of single qubit purification time at $p_c = 0.778$.

## 4.1 Cluster hybrid random circuits

As mentioned previously, in this case $M$ tunes the range of unitary gates, therefore higher value of $M$ leads to an extended scrambling of information in the unitary gate evolution layer.

**Critical exponents** A detailed comparison of $p_c$ with respect to $1/M$ is shown in Fig. 4 for CHRC. In Fig. 4, we observe a volume law to area law transition for $M = 2$ with $p_c = 0.321 \pm 0.001$, $\nu = 1.23 \pm 0.004$. The transition is conformal as signaled by $z = 1 \pm 0.001$ and by the absence of scaling dimension $\gamma_{\mathcal{Q}}$[3]. As $M$ is increased to $M = 4$, the $p_c$ shifts to a much larger value with $p_c = 0.778 \pm 0.001$, with $\nu = 1.37 \pm 0.042$ and $z = 1 \pm 0.001$. On further enlarging the range of unitary gates to $M = 8$ or $M = L$, we see that for system sizes under consideration $p_c$ approaches 1. In particular, for $M \geq 8$, we cannot distinguish within our errorbars whether the area law has a finite extent, or not.

In order to demonstrate the behavior of relevant observables in this case, we choose $M = 4$

---

[3]We preliminary considered a finite size scaling including also this term which gives results compatible to $\gamma_{\mathcal{Q}} = 0$. Hence we do not include this term in the finite size scaling of the CHRC.

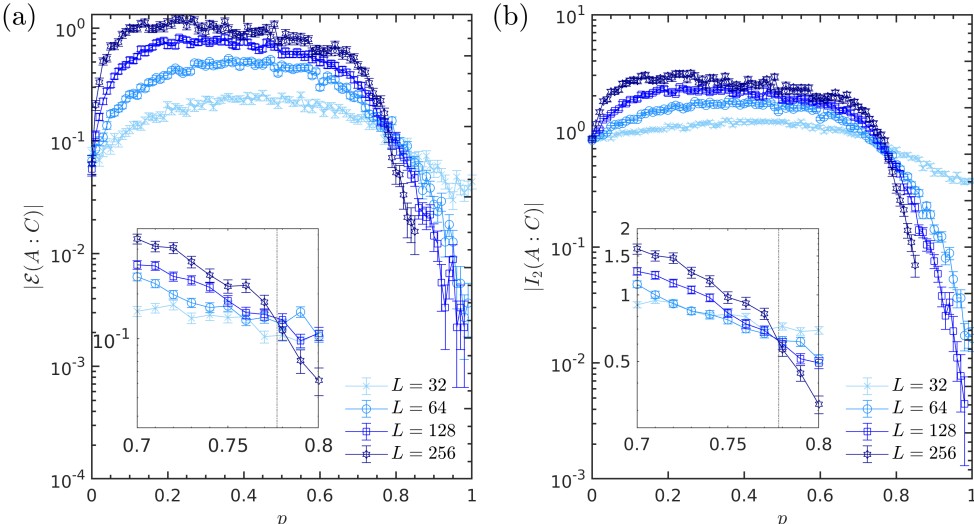

Figure 6: Comparison of negativity and MI as a function of $p$ for $M = 4$. (a) The behavior of negativity *w.r.t* $p$ shows a crossing at $p_c = 0.778$ for larger system sizes. (b) The plot of MI *versus* $p$ displays a similar behavior as that of negativity with crossing at the same $p_c$. The dashed-dotted line shows the crossing point $p_c$.

for illustration. The $M = 4$ value is specifically chosen owing to the fact that in this case the unitary gates are not just nearest-neighbour gates (arranged in ladder fashion as is the case for $M = 2$), but rather next to next nearest gates. This helps us probe the effect on the information propagation, i.e. $p_c$ and also on the universal exponents for circuits utilizing these finite ranged yet extended gates. The TMI as shown in Fig. 5 (a) for different system sizes shows a crossing at $p_c$: we note the data show a clear crossing behavior, which identifies a suitable region to perform a finite size scaling (FSS) analysis. To avoid spurious effect, we neglect the small system sizes ($L \leq 32$). The FSS collapse from Eq. (15) is obtained for $\nu = 1.37 \pm 0.042$ and $p_c = 0.778$, as shown in Fig. 5 (b). Similarly, the best collapse as obtained from this functional form for EE is for $\nu = 1.37$ and $p_c = 0.79$, which matched the exponent and critical $p$ as obtained from TMI within the error range. Lastly, we also draw out the dynamical critical exponent using Eq. (16). As shown in Fig. 5 (d), the dynamical exponent is $z = 1$ down to the percent level.

These values imply that, for the cases considered here where a transition at finite $p_c < 1$ is accessible, the exponents $\nu$ and $z$ appear to be in the universality of a short-ranged model [1, 2, 15], within error bars.

**Comparison: Entanglement negativity and Bipartite-Mutual Information**  A contrast between negativity and MI is also attempted in Fig. 6. Analogous to TMI, negativity and MI also show a crossing at $p_c = 0.778$ for larger system sizes. The smaller system sizes for the case of $I_2(A : C)$ (*e.g.* $L = 32$) in Fig. 6 (b) and $I_3(A : B : C)$ in Fig. 5 (a) visibly leads to an incorrect $p_c$ unlike in the case of $\mathcal{E}(A : C)$.

## 4.2 Long-range hybrid random circuits

For the long-range model our analysis is conveniently summarized in Fig. 7. In line with the consideration for the CHRC, we first analyze the stationary state, and once the critical point is identified within error bars, we extract the dynamical critical exponent from the ancilla qubit $S_R$.

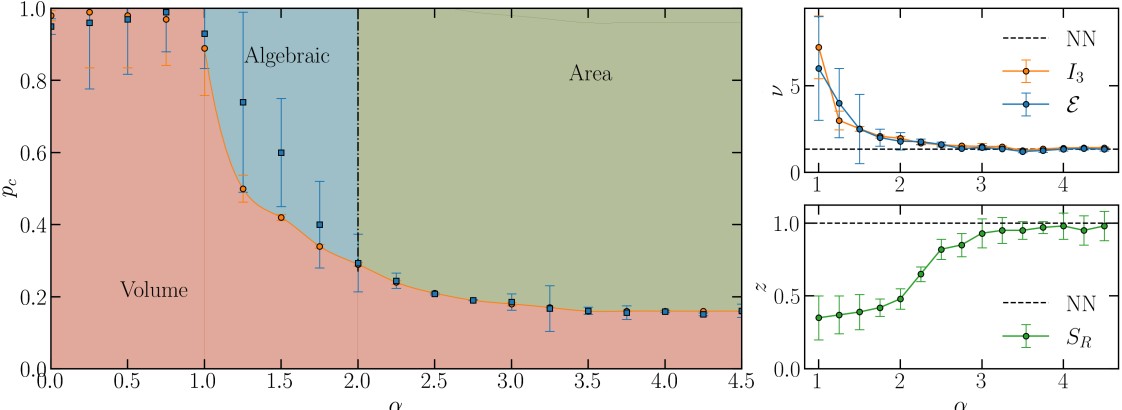

Figure 7: (Left) Phase diagram of the LRHRC. The analysis is performed with the TMI which presents sensibly less finite size effect compared to other quantities. The orange markers identifies the line of phase transitions $p_c(\alpha)$. For comparison we also present the estimated phase transition from the logarithmic negativity (blue markers). As clear from the error bars, the FSS suffers larger finite size effects compared to the TMI. Lastly, the crossover line at $\alpha = 2$ is obtained through the analysis of the negativity (cfr. Fig. 9) and by analytical arguments in the main text. (Right) Correlation length critical exponent as extracted from $I_3$ and $\mathcal{E}$, and dynamical critical exponent obtained through he analysis of $S_R$. For comparison, the dashed line NN, refers to the short-range case.

**Scaling of entanglement entropy and logarithmic negativity**   We first study the scaling of the half-system entanglement entropy and of the antipodal negativity (see Fig. 8 and Fig. 9, respectively). We find that for intermediate values of the parameter $\alpha$ these quantities develop an algebraic growth with system size.

To acquire insights, we consider the limit of high measurement rates $p \rightarrow 1$. Here the entanglement is solely captured by a single unitary gate, as at each timestep the state is reset to a random product state in the $Z$-basis. Then, the entanglement contribution is proportional to the average number of gates crossing the bipartition (for the entanglement entropy) or shared by the regions $A$ and $C$ (see Fig. 2) for the negativity. This computation is given in Ref. [51] for the entanglement entropy, which gives $S(L/2) \propto L^{2-\alpha}$ for $1 < \alpha < 2$ and $S(L/2) = \text{const}$ for $\alpha > 2$. For the negativity the computation is similar. The number of gates acting with one site in $A$ and the other in $C$ is given by

$$\mathcal{E}(A:C) \propto \int_{x \in A} dx \int_{y \in C} dy \frac{1}{|x-y|^\alpha} \propto L^{2-\alpha} + O(1), \qquad (17)$$

which implies that the negativity increases with system sizes for $\alpha < 2$, whereas it decreases for $\alpha > 2$. The subleading constant in Eq. (17) depends on the lattice spacing and other microphysical properties of the system.

These results qualitatively match the exponent extracted by the fit $\mathcal{E}(A:C) = aL^\kappa + b$, as shown in Fig. 9(Right). We attribute the discrepancies to finite size effects, which are more prominent at $\alpha < 1$ [4].

**Tripartite mutual information and entanglement negativity**   We obtain the phase diagram Fig. 7 by performing the finite size scaling on the tripartite mutual information and of the

---

[4]We do not report the same analysis on the entanglement entropy, which does not give additional information, but presents larger finite size effects

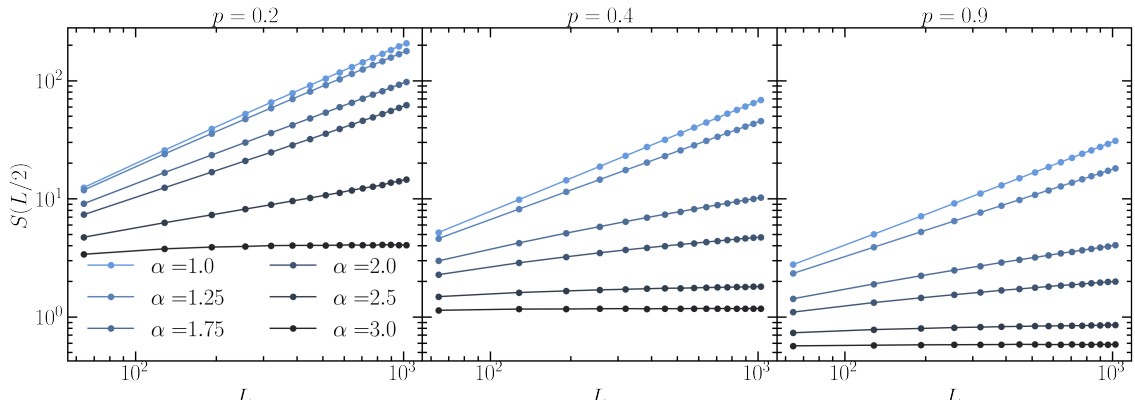

Figure 8: Scaling of the entanglement entropy for various interaction range $\alpha$ and various measurement rate $p$. We clearly see an algebraic scaling $S(L/2) \propto L^{\mu}$, with $\mu = 1$ signaling a volume law and $\mu < 1$ an algebraic law.

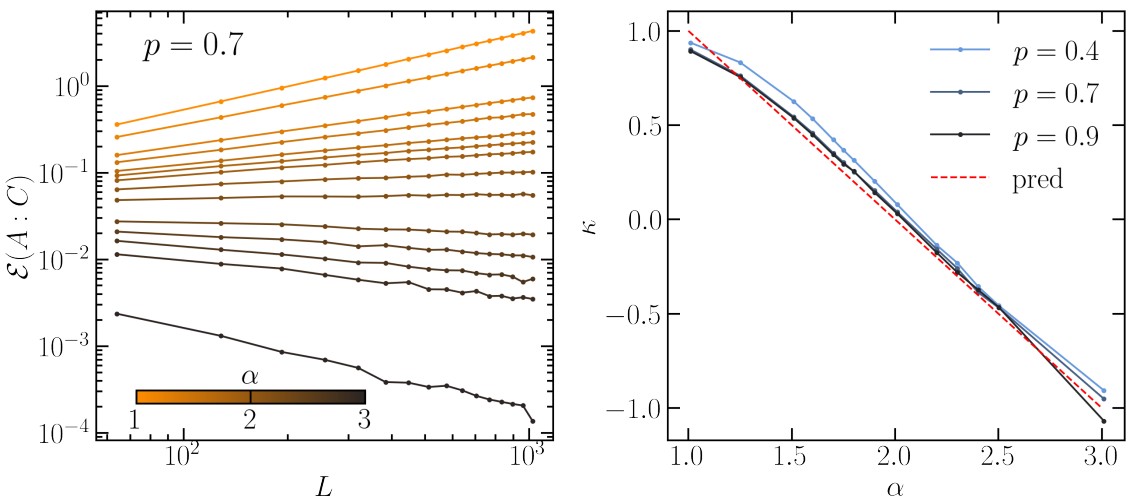

Figure 9: (Left) Scaling of the negativity for various interaction range $\alpha$ at high measurement rate $p = 0.7 > p_c$. (Right) Exponent $\kappa$ extracted by fitting $\mathcal{E}(A : C) = aL^{\kappa} + b$. Our results show a crossover between the region $\alpha \gtrsim 2$ for which the negativity decreases with system size, and $\alpha \lesssim 2$ where the negativity increases with system sizes. The prediction given via the counting argument Eq. (17), valid at $p \to 1$, is highlighted in red.

negativity, following the scaling hypothesis Eq. (15). As examples, we detail the scaling of TMI and negativity in Fig. 10 for $\alpha = 1.5$ and $\alpha = 3.5$. The data collapses (insets of Fig. 10) identify $\nu = 2.5(1)$, $\gamma_{I_3} = 0.3(1)$, $p_c = 0.41(1)$ and $\nu = 1.35(2)$, $\gamma_{I_3} = 0.02(2)$, $p_c = 0.160(2)$ for the TMI at, respectively, $\alpha = 1.5$ and $\alpha = 3.5$. Instead, for the negativity we have $\nu = 3(1)$, $\gamma_{\mathcal{E}} = 0.5(2)$, $p_c = 0.5(1)$ and $\nu = 1.30(2)$, $\gamma_{\mathcal{E}} = 0.01(1)$ and $p_c = 0.161(3)$ for, respectively, $\alpha = 1.5$ and $\alpha = 3.5$. Overall, our analysis shows compatibility between the critical points and the critical exponents found for the tripartite mutual information and the negativity, as show in Fig. 7. The summary of the scaling exponents $\gamma$ is given in Fig. 11. We see that for $\alpha \gtrsim 3$ the scaling dimension is compatible with a conformal field theory ($\gamma_{I_3}, \gamma_{\mathcal{E}} = 0$).

**Reference qubit** We extract the dynamical critical exponent $z$ by studying the behavior at $p = p_c$ of $S_R$ Eq. (16). The summary of the exponents is given in Fig. 7. Here we present

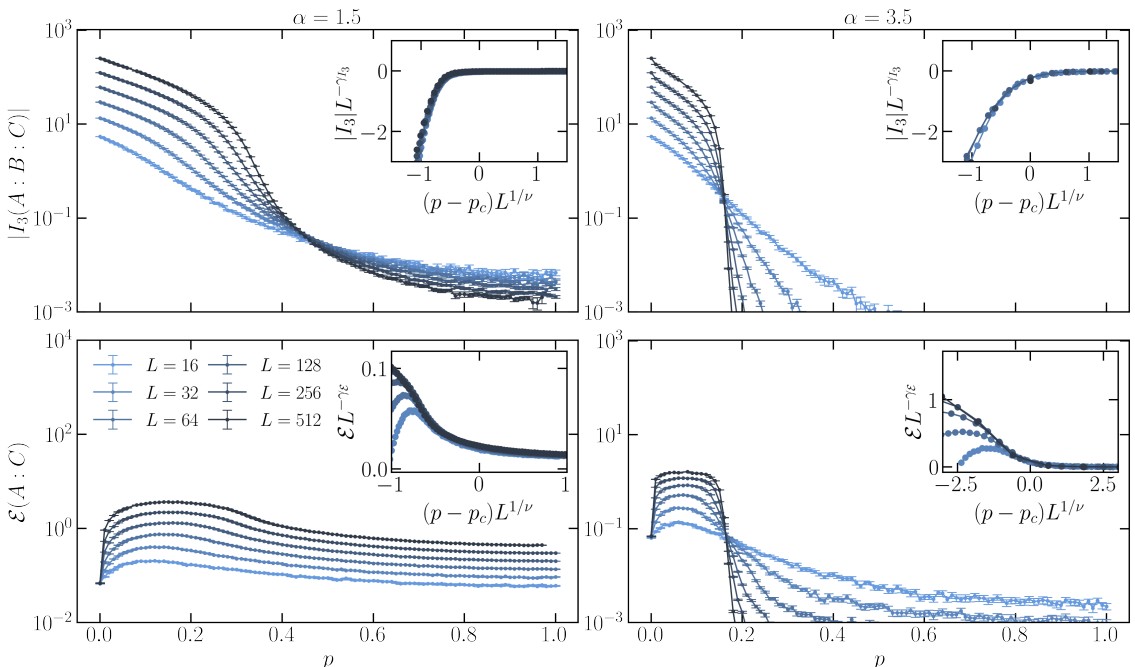

Figure 10: Data collapse for $\alpha = 1.5$ and $\alpha = 3.5$ of the tripartite mutual information (top panels) and the logarithmic negativity (bottom panels). For $I_3$, our FSS gives $\nu = 2.5(1)$, $\gamma_{I_3} = 0.3(1)$ and $p_c = 0.41(1)$ for $\alpha = 1.5$ and $\nu = 1.35(2)$, $\gamma_{I_3} = 0.02(2)$ and $p_c = 0.160(2)$. For the negativity we have $\nu = 3(1)$, $p_c = 0.5(1)$ and $\gamma_{\mathcal{E}} = 0.5(2)$ for $\alpha = 1.5$ and $\nu = 1.30(2)$, $\gamma_{\mathcal{E}} = 0.01(1)$ and $p_c = 0.161(3)$.

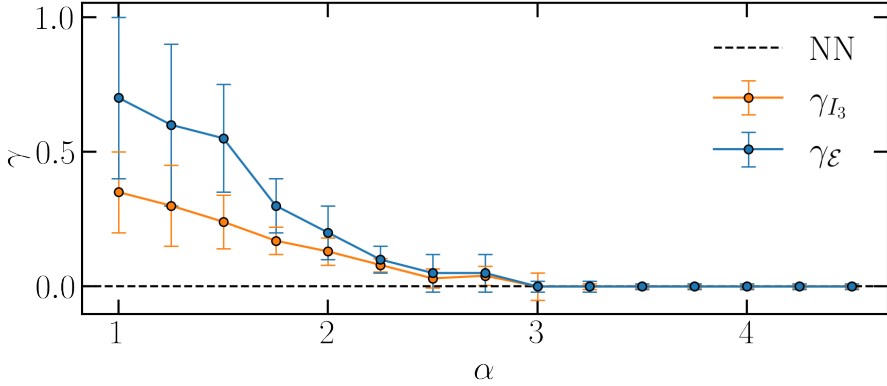

Figure 11: Scaling dimension of the logarithmic negativity and of the tripartite mutual information.

some samples of the data collapses for few values of the parameter $\alpha = 1.5, 2.5, 4$, finding respectively $z = 0.33(7)$, $z = 0.81(4)$ and $z = 0.95(8)$ (see Fig. 12). For $\alpha \gtrsim 3$ the dynamical exponent is compatible with the one of a conformal field theory, which require spacetime isotropy $z = 1$.

## 5 Discussion and Conclusion

Using the information theoretic measures like entanglement entropy, mutual information, entanglement negativity, as well as protocols involving an ancillary spin, we have numerically

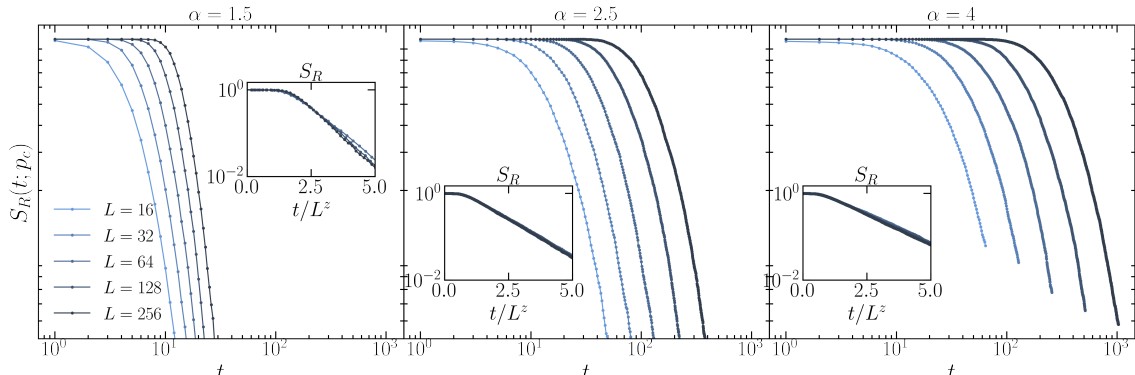

Figure 12: Dynamical scaling of $S_R$ for $p_c = 0.41$ (Left), $p_c = 0.22$ (Center), and $p_c = 0.16$ (Right). The collapse is obtained respectively for $z = 0.33(7)$, $z = 0.81(4)$, and $z = 0.95(8)$.

Table 1: Summary of the results for the CHRC.

| CHRC | | | | |
|---|---|---|---|---|
| | **Transition** | $\mathbf{p_c}$ | $\boldsymbol{\nu}$ | **z** |
| **M = 2** | Vol.-Area | $0.327 \pm 0.001$ | $1.276 \pm 0.114$ | $1 \pm 0.01$ |
| **M = 4** | Vol.-Area | $0.78 \pm 0.01$ | $1.37 \pm 0.102$ | $1 \pm 0.01$ |
| **8 ≤ M ≤ L** | Vol. | $\to 1$ | $-$ | $-$ |

Table 2: Summary of the results for the LRHRC.

| LRHRC | | | | | | | |
|---|---|---|---|---|---|---|---|
| | **Transition** | $\mathbf{p_c}$ | $\boldsymbol{\nu}$ | **z** | $\boldsymbol{\gamma_{I_3}}$ | $\boldsymbol{\gamma_{\mathcal{E}}}$ | **Class** |
| $\mathbf{0 \leq \alpha \leq 1}$ | Vol. | $p \to 1$ | - | - | - | - | - |
| $\mathbf{1 < \alpha \leq 2}$ | Vol.-Algeb. | $1 \div 0.28$ | $5 \div 1.5$ | $0.35 \div 0.5$ | $0.4 \div 0.15$ | $0.6 \div 0.25$ | Non-CFT |
| $\mathbf{2 < \alpha < 3}$ | Vol.-Area | $0.28 \div 0.17$ | $1.5 \div 1.35$ | $0.5 \div 1$ | $0.15 \div 0$ | $0.25 \div 0$ | Non-CFT |
| $\boldsymbol{\alpha \geq 3}$ | Vol.-Area | $p_c \to 0.16$ | $\nu = 1.35$ | $1$ | $0$ | $0$ | CFT |

investigated measurement induced phase transitions in two protocols involving non-local, two-qubit Clifford gates. We summarize our main results in Tables 1 (CHRC) and 2 (LRHRC). For the case of CHRC, the universal properties associated with the transition are compatible with short-range Clifford circuits, whereas the extent of the volume law (error-correcting) region changes significantly. We note that a similar observation has been reported in a recent analysis of fast-scrambler circuits in Ref. [86]. Here the authors study a similar model of finite range random circuits – with two-body random unitary gates acting on spins distant $k$ sites, and find a transition between a volume law and an area law sharing the same fixed point as that of short-ranged Clifford circuits. Importantly, the critical measurement rate $p_c$ increases with the spin separation $k$.

A remarkable result is that for finite $M \simeq 8$ and above, our data cannot distinguish between a $p_c < 1$ and $p_c = 1$, the latter corresponding to no measurement-induced transition in the thermodynamic limit.

For the case of LRHRC we find a rich phase diagram, where the power-law distributed unitary interactions are leading actors in determining the properties of the stationary phase. For $\alpha < 1$ the system persists in a volume-law phase for any measurement rate $p$. In this regime, each unitary layer act as a global random gate on the full Hilbert space, hence at each time step the state points toward a fully random stabilizer state. Instead for $\alpha \in [1, 3]$ we iden-

tify a line of non-conformal critical points separating a volume-law phase from a phase with subextensive system size scaling of the entanglement entropy. The subextensive phase exhibits algebraic growth of the entanglement entropy with $L$ for $1 < \alpha < 2$ and a constant value for $\alpha > 2$ (area law). This separation is neat in the antipodal negativity, which increases algebraically for $1 < \alpha < 2$ and decreases algebraically for $\alpha > 2$. We locate the crossover point $\alpha = 2$ using a counting argument in the limit $p \to 1$. The numerics for $p_c < p < 1$ suggests the qualitative properties of the phase are captured with the limit $p \to 1$. In particular, the algebraic exponent characterizing the system size scaling of the negativity ($\kappa$) matches the value obtained from the counting argument $\kappa_{\text{count}} = 2 - \alpha$. Finally, for $\alpha \geq 3$ our findings conclude the phase transition is between a volume-law and an area-law phase, with the universality class compatible with that of short range clifford circuits [2]. Overall, parts of our analysis are compatible with the results obtained in Ref. [51], where a similar but different long-range protocol is considered, suggesting these models critical point belong to the same $\alpha$-dependent universality classes. Another recent article [87] considering tractable large-N models also investigated the effects of power-law long-range couplings on measurement induced phase transitions to analytically derive the phase diagram and probe the critical exponents. Some of the critical features we discussed are present in their treatment as well, further suggesting the generic features of those. Analytical results on the rich phenomenology highlighted by our simulations on random Clifford circuits, and supported by the study of similar models [51,86], are desirable and left for future work. We stress that the analytical arguments on long-range and finite range model are at present phenomenological pictures [7,51] with very good numerical match. An ab inition understanding of long-range Clifford circuits would require, e.g., extending the considerations in Ref. [88] to the context of variable-range hybrid random circuits (since it is by now understood that, at least for spin-1/2 systems, circuits generated by Clifford unitaries behave rather differently from the ones generated by Haar unitaries). The non-trivial nature of the problem can be understood in terms of the mapping between random circuits and classical statistical mechanics models [59,60,89]. The resulting lattice, both for the cluster and long-range models considered in this work, features a convoluted geometry, which stem from the interaction range, and which does not simplify in the standard limit of infinite on-site Hilbert space dimension.

## Acknowledgements

We thank G. Pagano for discussions. This work is partly supported by the ERC under grant number 758329 (AGEnTh), by the MIUR Programme FARE (MEPH), by a Google Quantum Research Award, and by European Union's Horizon 2020 research and innovation programme under grant agreement No 817482 (Pasquans). XT is supported by the ANR grant "NonEQuMat" (ANR-19-CE47-0001). We acknowledge computational resources on the Collége de France IPH cluster.

## A   Stabilizer formalism and Clifford hybrid evolution

A stabilizer state $|\psi\rangle$ for $L$ qubit is defined as a state for which there exists $L$ independent non-trivial Pauli strings $O_i$ such that $O_i|\psi\rangle = |\psi\rangle$. Clearly, every pair of operators $O_i$ fulfilling this property commute, hence they form an Abelian group. We denote the group generated by the $\{O_i\}$ as the stabilizer group $\mathcal{G}$ (for the state $|\psi\rangle$).

The stabilizer group uniquely define the state $|\psi\rangle$. Given a Pauli string

$$O = e^{i\pi\phi} X_1^{n_1} Z_1^{m_1} X_2^{n_2} Z_2^{m_2} \cdots X_L^{n_L} Z_L^{m_L} , \tag{18}$$

it follows that $(1+O)/2$ is the projector on the $+1$ state. Hence, for a stabilizer state we have

$$|\psi\rangle\langle\psi| = \prod_{i=1}^{L}\frac{1+O_i}{2} = \frac{1}{2^L}\sum_{O\in\mathcal{G}}O\,. \tag{19}$$

From the relation Eq. (19) we conclude that it is equivalent to consider the group $\mathcal{G}$ or the state $|\psi\rangle$ [64, 66]. In particular, the full group can be encoded in a $(2L+1)\times L$ matrix $\mathbb{G} = [\phi^j, n_i^j, m_i^j]$, with elements in the field $\mathbb{Z}_2$, where each row encode the Pauli string exponents Eq. (18) for the generators $O_i$ [5]. With the above prescription, the complexity is polynomial, provided the calculations can be performed directly acting on $\mathbb{G}$ (denoted symplectic representation). This is the case for the hybrid random evolution and for the computation of the entanglement measures we perform in this work.

## A.1 Hybrid quantum evolution: review of the Gottesman-Knill theorem

The Gottesman-Knill theorem [64–66] give operational instructions on how to implement Clifford unitary gates and projective measurement directly on the symplectic representation $\mathbb{G}$. In this subsection we sketch this result.

First, we recall that Clifford unitary gates by definition map a single Pauli string to a single Pauli string. Given the Clifford unitary $U$ acting on the Hilbert space, we denote $u$ its $2L+1\times 2L+1$ matrix representation which acts on the symplectic representation of stabilizer states. Then the state $|\psi'\rangle = U|\psi\rangle$ is encoded in the stabilizer group $\mathcal{G}'$ encoded in the matrix $\mathbb{G}' = u\mathbb{G}$ (where the matrix-matrix multiplication is in the field $\mathbb{Z}_2$).

Similarly, projective measurement on Pauli strings (as the ones considered in this work) preserve the symplectic representation. Suppose we wish to project onto $(1+O_p)/2$. We preliminary organize the generators in such a way that for $i < k$, $[O_i, O_p] = 0$, and $k \le i \le L$, $\{O_i, O_p\}$ (each generator $O_i$ either commute or anticommute with $O_p$, provided the $\{O_i\}$ and $O_p$ are all Pauli strings). After the measurement, the state include the projection $(1+O_p)/2$. Then, the remaining operators are reorganized in a way that the final generators commute. This can be achieved in multiple way (there is a gauge freedom in the choice of generators); for instance we notice that the product of two operators anticommuting with $O_p$ commute with $O_p$. Then $\mathcal{G}' = \text{span}(O_1, \ldots, O_{k-1}, O_p, O_k O_{k+1}, \ldots, O_k O_L)$ describes the state $|\psi'\rangle = (1+O_p)|\psi\rangle/2$. This prescription can be easily encoded in the symplectic formalism (see Ref. [65] for details, also on how the readout of a measurement of $O_p$ are obtained).

We conclude by a remark. For our purposes, the phase does not play any role, as the entanglement entropy and negativity are unaffected by the phases of the stabilizers. (*En passant*, this allows neglecting the measurement result readout). Hence, in actual computations, we drop the phase column, and consider the restricted $2L \times L$ matrix $\tilde{\mathbb{G}}$ obtained neglecting the $\phi_j$.

## A.2 Computation of entanglement entropy and negativity

In this subsection we briefly review results in Ref. [67, 68] concerning the entanglement entropy, and in Ref. [17, 18] concerning the negativity.

**Entanglement entropy** Given a stabilizer state and a bipartition $A \cup B$, the entanglement entropy is given by the $2L_A \times L$ matrix $\mathbb{G}_A$ obtained restricting the site indices in $\tilde{\mathbb{G}}$ on the subsystem $A$.

$$S_A = \text{rank}_{\mathbb{Z}_2}(\mathbb{G}_A) - L_A\,, \tag{20}$$

---

[5]The phase $\phi$ for a stabilizer can only be 0 or 1. See Ref. [64] for a detailed discussion.

where $L_A$ is length of subsystem $A$ and the rank is obtained using Gaussian elimination to calculate the reduced echelon form of a matrix with modulo 2.

**Entanglement negativity**    We consider a stabilizer state $|\psi\rangle$ encoded in $\mathcal{G}$ and a tripartition $A \cup B \cup C$. Tracing out the subsystem $C$, we get

$$\rho_{AB} = \mathrm{tr}_C |\psi\rangle\langle\psi| = \frac{|\mathcal{G}_{AB}|}{2^{L_{AB}}} \left( \frac{1}{|\mathcal{G}_{AB}|} \sum_{O_{AB} \in \mathcal{G}_{AB}} O_{AB} \right), \tag{21}$$

where we have defined the subgroup

$$\mathcal{G}_{AB} = \{O_{AB} | O_{AB} \otimes \mathbf{1}_C \in \mathcal{G}\}. \tag{22}$$

We note that this subgroup can also be trivial, and Eq. (21) is a consequence of the traceless property of the Pauli strings. Hence, the partial trace on $C$ discriminate all the group elements with non-trivial action on $C$. We define the $m \times m$ matrix $J$ such that

$$J_{ij} = \begin{cases} 1 & \text{if } \{O_A^i, O_A^j\}, \\ 0 & \text{otherwise}, \end{cases} \tag{23}$$

where we split each $O_{AB}$ as $O_{AB} = O_A \otimes O_B$. Then the following holds

$$\mathcal{E}(\rho_{AB}) = \frac{1}{2} \mathrm{rank}_{\mathbb{Z}_2} J. \tag{24}$$

We refer to Ref. [17,18] for a throughout analysis and a proof of Eq. (24).

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
