# Peer review of "Measurement-induced criticality in extended and long-range unitary circuits"

_SciPost Physics Core, doi:SciPost Phys. Core 5, 023 (2022)_

## Round 2 · Referee Report · Anonymous (Referee 1) · 2021-11-30

Strengths

1) relevant
2) thoroughly investigated
3) clearly communicated

Report

The paper investigates random circuits, where the applied quantum gates are randomly drawn from the Clifford group and where Pauli measurements are randomly performed in between consecutive gates . This permits a classical investigation of the dynamics of the models. Phase transitions in the dynamical evolution are explored as the range of interactions (implemented by the chosen gates) is varied. The paper predicts a range of interesting observations as the behavior of the critical exponents is studied. While not unexpected, a quantitative investigation of the findings is a highly relevant contribution to the very active field of dynamics in random circuits. Moreover the investigations are likely to become relevant for possible future experiments provided sufficient qubit numbers are reached.

  • validity: top
  • significance: high
  • originality: good
  • clarity: top
  • formatting: perfect
  • grammar: excellent

Author:  Marcello Dalmonte  on 2022-01-19  [id 2111]

(in reply to Report 1 on 2021-11-30)
Category:
answer to question

We thank Referee 1 for their reading of our manuscript, and their appreciative comments.

---

## Round 2 · Referee Report · Anonymous (Referee 2) · 2021-12-15

Strengths

1.) high relevance
2.) well structured presentation
3.) thorough analysis

Report

The authors study two instances of long-range hybrid random Clifford circuits. This permits effective classical simulations and various phase transitions are studied in this setting.
In case of circuits where the probability of long-range two-qubit gates decays algebraically they confirm and extend the results found in [1].
The authors furthermore study multi-qubit (clustered) interactions which, to my knowledge, is the first time this has been done explicitly. They find that when the clusters are below a certain size usual measurement induced criticality is found, while in the limit of large clusters scrambling always wins.

The results found in this paper corroborate the general picture of long-range interacting hybrid circuits outlined in [1] and [2,3] for free and interacting fermions respectively. The study of hybrid multi-qubit interactions goes well beyond setups studied so far and is therefore a relevant contribution to the newly emerging field of hybrid dynamics.

Due to the thorough and timely analysis I recommend this paper for publication.

[1] https://arxiv.org/abs/2104.13372
[2] https://arxiv.org/abs/2105.08076
[3] https://arxiv.org/abs/2104.09118

suggestion:
Consider changing Fig. 8 to log-log plot (L or 1/L) to highlight algebraic scaling of entanglement entropy.

typos:
1.) p. 2 "... Clifford circuits provide a viable path to understand ..."
2.) p. 22 Reference [50] "T. Müller ..."

  • validity: top
  • significance: high
  • originality: good
  • clarity: high
  • formatting: perfect
  • grammar: excellent

Author:  Marcello Dalmonte  on 2022-01-19  [id 2110]

(in reply to Report 2 on 2021-12-15)
Category:
answer to question

We thank Referee 2 for their careful reading, and the comments. We have changed the scales of Fig. 8 as per their suggestion, and corrected several typos in the text, including those pointed out in the report.

---

## Round 2 · Referee Report · Anonymous (Referee 3) · 2021-12-21

Strengths

1- The topic is very relevant and the numerical investigation of the two-different long-range circuit models is comprehensive to a set of other recently appeared long-range models.
2- The paper is very accessible and provides a short and crisp summary of the obtained results. I find it particularly well written.
3- The summary of the entanglement-based observables and the numerical procedure are very precise and helpful.

Weaknesses

1- While I find it helpful for the reader to display a Hamiltonian, which corresponds to the circuit dynamics, the precise connection between the two Hamiltonians and the circuit dynamics is missing, and might therefore be misleading.
2- The manuscript presents an exhaustive list of results but falls short in providing a deeper discussion or conclusion about the physics in the long-range model. After reading the paper, it feels like a list of numbers and facts but I'm not sure if I have gained much deeper insights.

Report

The manuscript by Sharma et al. represents a comprehensive examination of long-range unitary circuit dynamics subject to local measurements. It appears very timely, since the entanglement dynamics in long-range hybrid quantum circuits has attracted a lot of attention recently. The scientific evaluation and presentation is very accessible and of high standard. Overall, I support publication of the present manuscript in SciPost Physics Core.

Below I mention two recommendation for the authors for potential changes. I'd leave it up to them if they want to consider them or not.

Requested changes

1- The authors might want to consider the points I mention under weaknesses. Especially a rectification of the correspondence to a Hamiltonian evolution might be very helpful and in order. While there is probably an analogy to the Hamiltonian models, it is clear that the circuits cannot mimic those Hamiltonians. Any continuous evolution with the interacting Hamiltonians leaves the space of free wave functions (either stabilizers or free fermions/bosons) and thus corresponds to a different class of correlated many-body states.
2- A little bit more of discussion of the results might be helpful. For instance some understanding of the the long-range models (also in terms of entanglement generation) was provided in the Refs. 49-51 and it might not be the worst to repeat some aspects of this discussion very briefly.

  • validity: high
  • significance: good
  • originality: good
  • clarity: top
  • formatting: excellent
  • grammar: excellent

Author:  Marcello Dalmonte  on 2022-01-19  [id 2112]

(in reply to Report 3 on 2021-12-21)
Category:
answer to question

We thank Referee 3 for their reading of our manuscript, and their comments. We have considered both of them in our revision of the work, as follows:

1) we have changed considerably the text in correspondence to the parallelism, as we have noted this generates confusion in the interpretation of the protocols;

2) while of course the predictions of Ref. [49-51] are important (and also supported by checks therein), it is not fully clear to us to which extent those can be applied to the present settings. A more refined method, based on the recently introduced effective statistical mechanics models for Clifford circuits in 2110.02988, seems promising in this direction. While we feel this goes well beyond the scope of the present work, we have revised the conclusions to emphasize this perspective.

---

## Round 2 · Referee Report · Anonymous (Referee 4) · 2021-12-22

Strengths

1 - Timeliness
2 - Fresh and fruitful idea to compare the information dynamics in quantum circuits with clustered and long-range unitary gates
3 - Several complementary observables are studied numerically

Weaknesses

1 - The models studied numerically are introduced and described in not particularly clear terms
2 - The manuscript presents numerical studies only, without any attempts of describing and explaining the results in analytical terms (within some toy model, or, at least, by means of plausible speculations)
3 - The current version of the manuscript only reports observations; the work would highly benefit from strengthening the conclusions based on these observations and highlighting the relevance of these conclusions to the field

Report

The manuscript addresses numerically entanglement in the two specific models of quantum circuits with spatial correlations. The subject of the study is very topical; the obtained results, showing the difference in the information dynamics and the entanglement phase diagrams between the two models, are novel and interesting.

Unfortunately, the authors suggest no general conclusions based on their numerical results, which could emphasize the relevance of the work to the field. The statements like "... the fi xed point is compatible with the one of short-range Clifford circuits" and "... consistent with the one reported in Ref. [51], suggesting that ... the measurement induced transition is governed by the same underlying ... theory despite the protocols being different" are too specific to the models addressed in the paper. Apparently, these models were chosen to demonstrate some more general properties of the corresponding classes of systems. For this purpose, a more extended discussion of the numerical observations in analytical terms is necessary.

The description of the models and the protocol should be modified. The main problem is that the unitary gate defined in Eq. (3) does not seem to define the CHRC as described in Fig. 1. Indeed, if one naturally assumes that Eq. (3) describes each of U-blocks in Fig. 1a, i.e., the time variable t corresponds to the whole block there, the "fine structure" (internal architecture) of the large U-block shown in Fig. 1b (left) is not actually defined in Eq. (3). Indeed, the pattern of the elementary small boxes in Fig. 1b (left) is very specific, suggesting the introduction of a certain "time-ordering" inside the large U-block, which is missing in the formal equation for U(t). ("The unitary gates U(t) are laid out in a manner that mimics a soft-shoulder potential extending over M sites" -- is this layout crucial for the CHRC model, or one can use an arbitrary ordering?). If, however, the time step t refers to a horizontal slice in Fig. 1b(left) rather than in Fig. 1a, Eq. (3) does not describe Fig. 1b(left) since all sites i are involved in the product in Eq. (3), while there are empty sites in each elementary slice in Fig. 1b(left). To avoid confusion, the authors should define their models (and notation) in a much more careful way!

The randomness in the gates, as well as the definition of averaging over this type of randomness, should also be described explicitly in analytical terms.

In addition to this major point, there are also related minor issues:
- The circuit realization Km in Eq. (1) requires an explicit analytical representation in the form Km=... (may be given after defining the elements of the hybrid circuit)
- It would be nice to add the label (+-) to the projectors in Eq. (2)
- The caption of Fig. 1 says: "M = 4 in the above illustration". However, this is not what is seen in the figure, where the dashed links between the largest pairs of the smallest blocks have different lengths (in particular, the top left block obviously has M>4)
- The different colors used for pairs in Fig. 1b(right) should be explained in the figure caption

Now, turning to the results, it would be nice to have the data points for M=3 and M=5 in Fig. 4, especially in the upper right panel for nu (by the way, the panels are not labeled in contrast to what the figure caption says). Is it clear why M=6 is the "magic cluster range" beyond which the area-law phase disappears?

Figure 7: It is not clear what the numerical evidence for the vertical dashed-dotted line at alpha=2 is provided. Is it at all possible, with available system sizes and accuracy shown by error bars, to distinguish in such a phase diagram between the "algebraic phase" and a crossover region separating the area-law and volume-law phases?

Finally, it would be interesting to compare the findings

In conclusion, while the manuscript present interesting new numerical results, I cannot recommend it for publication in its present form (see above). The models should be described in a more careful way, and some numerical observations should be further clarified. The manuscript can be reconsidered for publication after the authors have modified the manuscript accordingly. In addition, I strongly suggest that the authors extend the discussion section (which is currently only presents a list of observations) by adding some analytical arguments emphasizing the generality of their findings. Recent preprints https://arxiv.org/abs/2110.02988 and https://arxiv.org/abs/2111.08018 might be helpful in this respect.

PS. Minor grammar issues/typos (an incomplete list):

p. 2: "These systems comprise of random unitary gates..." --
perhaps, the authors meant "consist of" or "comprise" (without "of"); for a related discussion on "comprised of", see
https://en.wikipedia.org/wiki/User:Giraffedata/comprised_of
https://www.theguardian.com/commentisfree/2015/feb/05/why-wikipedias-grammar-vigilante-is-wrong

p. 2: "where analytical results where obtained" -- were obtained

p. 3: "Our analysis suggest the fixed point" -- suggests (or "our analyses suggest", but this sounds weird)

p. 4: "the system belong to the same universality" -- belongs

p. 5: "A review of stabilizer formalism, Clifford group and on the efficient numerical implementation based on the Gottesmann-Knill theorem are detailed in the Appendix" -- please, reconsider the sentence

p. 6, figure caption: "The normalization of the probability distribution P(r) impose..." -- imposes

p. 8: "We remark that SR act as an order parameter..." -- acts

p. 13: "Overall, our analysis show compatibility..." -- shows

p. 15: "Another recent article [90] considering tractable
large-N models also investigate..." -- investigated

Requested changes

1 - Clarify the model in Sec. 2

2 - Respond to question concerning Figs. 4 and 7 from the report

3 - Strengthen the discussion of the observations, putting the findings in a more general context and adding (semi-)analytical arguments

4 - Correct typos

  • validity: good
  • significance: good
  • originality: high
  • clarity: ok
  • formatting: good
  • grammar: reasonable

Author:  Marcello Dalmonte  on 2022-01-19  [id 2113]

(in reply to Report 4 on 2021-12-22)

See attached pdf file.

Attachment:

ReplyRef4.pdf

---

## Round 4 · Referee Report · Anonymous (Referee 3) · 2022-1-20

Strengths

1- The topic is very relevant and the numerical investigation of the two-different long-range circuit models is comprehensive to a set of other recently appeared long-range models. 2- The paper is very accessible and provides a short and crisp summary of the obtained results. I find it particularly well written. 3- The summary of the entanglement-based observables and the numerical procedure are very precise and helpful.

Weaknesses

1- After the revision by the authors, the paper still has a strong focus on the numerical results and less on providing a physical picture of the underlying dynamics.

Report

The authors have considered the suggestions from my previous report and revised the manuscript accordingly. While I agree that an analytical understanding of the presented results is probably not straightforwardly obtained, I also think it is a chance missed by the authors to significantly improve their work and the impact it will have in the future.
However, there are plenty of new and interesting results on the topic of measurement-induced phase transitions presented in the manuscript. It is well written and very accessible and I believe it is a good paper that should be published in SciPost Physics Core.

---

## Round 4 · Referee Report · Anonymous (Referee 4) · 2022-2-6

Strengths

same as in my previous report

Weaknesses

1 - The description of the CHRC model in Sec. 2.1 is still somewhat misleading

Report

The authors have done a good job addressing the referees' remarks from the first round. However, the formal description of the CHRC model in Sec. 2.1 remains a problem, despite the authors' efforts to improve it. Indeed, Eq. (3) appears to be a product of elementary two-body gates U_i,i+r,t. Here, t is the elementary time-step (the vertical size of the smallest rectangular element in the left panel of Fig. 1b). On the other hand, the first sentence of Sec. 2.1 says: "The unitary evolution is a sequence of M-body cluster unitary gates Ui,j,t, each of which is build stacking two body unitary gates". This implies that Ui,j,t is a composite (cluster) gate comprising three two-body elementary gates in Fig. 1b, so that time step t is the vertical size of such a cluster, which is 3 elementary layers in the figure. If I use this definition of the quantity Ui,j,t in Eq. (3), I would get a nonsensical result. Let us, therefore, assume that Ui,j,t is a two-body gate. Figure 1b suggests that, with this definition, at each time t there is only one two-body gate. On the contrary, Eq. (3) tells us that at each time t we should multiply many two-body gates. In particular, for L=6 (with periodic boundary conditions) and M=4, as in Fig. 1b, one gets the following product, if one literally follows the prescription of Eq. (3): U_1,2,t*U_1,3,t*U_1,4,t*U_2,3,t*U_2,4,t*U_2,5,t*... This is not what I see in Fig. 1b, where, for example, for the bottom layer (t=1) I see only U_1,2,1, while all other two-body gates in this product are clearly trivial, i.e., U=1. Note that in the product of two-body operators, as written in Eq. (3), again, the ordering of operators is not specified explicitly, in contrast to what is written in the footnote. In order to have a formal correspondence between Eq. (3) and Fig. 1b, the authors should (i) modify the first sentence of Sec. 2.1; (ii) write explicitly that, for a given value of t, only a single factor in the product in Eq. (3) is not equal to unity in the example of Fig. 1b (this would automatically resolve the problem of the operators' ordering); (iii) to give an explicit expression (perhaps, in terms of a Kronecker delta-symbol involving i, r, and t) specifying the indexes of the non-trivial two-body gate in the product (3) for the shoulder-like arrangement of Fig. 1b.

Requested changes

Improve the mathematical definition of the CHRC model

  • validity: high
  • significance: good
  • originality: high
  • clarity: good
  • formatting: excellent
  • grammar: excellent

Author:  Marcello Dalmonte  on 2022-03-19  [id 2304]

(in reply to Report 2 on 2022-02-06)
Category:
answer to question

We thank the Referee of their reading of our revised version. We have changed the text before and after Eq. 3, to accommodate extra explanations following the points in the report. We have also revised Fig. 1b caption accordingly."

---

## Round 4 · Referee Report · Anonymous (Referee 2) · 2022-3-15

Report

The authors have included all the requested changes of the referees (especially report 4) and I stand by my previous recommendation that this paper is a worthy contribution to scipost physics core.
The paper provides thorough numerical study of long range Clifford dynamics under measurement. While I partly agree that a stronger statement could be made in conjunction with analytics, I also believe that this is a highly non-trivial endeavor and well beyond the scope and intention of this article.

---

## Round 4 · Author Response

Dear Editor,

We thank the Referees for their careful reading of our manuscript and for their constructive comments and criticisms.
We have embodied these discussions in the new version of the manuscript, which we now hope is suitable for publication in Scipost Physics Core.

---

## Round 4 · List of Changes

In response of the Referee reports, we have implemented the following changes: - Corrected typos; - Added Ref [90] in bibliography; - Changes in Sec. 2 clarifying the models and the difference with Hamiltonian setups; - Updated caption of Fig 1; - Replaced linear scale with logarithmic scale in the x-axis of Fig 8; - Expanded discussion in Sec 5 .

---

## Round 5 · Author Response

Dear Editor,

we thank you for handling our work, and the Referees for the additional comments.

We have amended the description around Fig. 1b for clarity.

Yours sincerely

the authors

---

## Round 5 · List of Changes

1) Explicitly wrote down the layer as a product of clusters U_{{I,\dots,i+M-1},t}
2) Explicitly wrote the unitary gates within a single cluster
3) Adapted the figure captions.

---

## Editorial Decision

published